# Wrinkling on Stimuli-Responsive Functional Polymer Surfaces as a Promising Strategy for the Preparation of Effective Antibacterial/Antibiofouling Surfaces

**DOI:** 10.3390/polym13234262

**Published:** 2021-12-06

**Authors:** Carmen M. González-Henríquez, Fernando E. Rodríguez-Umanzor, Matías N. Alegría-Gómez, Claudio A. Terraza-Inostroza, Enrique Martínez-Campos, Raquel Cue-López, Mauricio A. Sarabia-Vallejos, Claudio García-Herrera, Juan Rodríguez-Hernández

**Affiliations:** 1Departamento de Química, Facultad de Ciencias Naturales, Matemáticas y del Medio Ambiente, Universidad Tecnológica Metropolitana, Santiago 7800003, Chile; guezu@utem.cl (F.E.R.-U.); matias.alegriag@utem.cl (M.N.A.-G.); 2Programa Institucional de Fomento a la Investigación, Desarrollo e Innovación, Universidad Tecnológica Metropolitana, Santiago 8940000, Chile; 3Programa PhD en Ciencia de Materiales e Ingeniería de Procesos, Universidad Tecnológica Metropolitana, Santiago 8940000, Chile; 4Research Laboratory for Organic Polymer (RLOP), Facultad de Química y Farmacia, Pontificia Universidad Católica de Chile, Santiago 7810000, Chile; cterraza@uc.cl; 5Group of Organic Synthesis and Bioevaluation, Instituto Pluridisciplinar, Universidad Complutense de Madrid, Associated Unit to the ICTP-IQM-CSIC, 28040 Madrid, Spain; emartinezcampos79@gmail.com (E.M.-C.); raquelcuelopez1x@gmail.com (R.C.-L.); 6Departamento de Ingeniería Mecánica, Facultad de Ingeniería, Universidad de Santiago de Chile, Santiago 9170022, Chile; mauricio.sarabia@usach.cl (M.A.S.-V.); Claudio.garcia@usach.cl (C.G.-H.); 7Polymer Functionalization Group, Departamento de Química Macromolecular Aplicada, Instituto de Ciencia y Tecnología de Polímeros-Consejo Superior de Investigaciones Científicas (ICTP-CSIC), 28006 Madrid, Spain; jrodriguez@ictp.csic.es

**Keywords:** pH-sensitive hydrogel, poly(acrylic acid), quartz crystal microbalance (QCM), ellipsometric measurement, antibacterial activity

## Abstract

Biocompatible smart interfaces play a crucial role in biomedical or tissue engineering applications, where their ability to actively change their conformation or physico-chemical properties permits finely tuning their surface attributes. Polyelectrolytes, such as acrylic acid, are a particular type of smart polymers that present pH responsiveness. This work aims to fabricate stable hydrogel films with reversible pH responsiveness that could spontaneously form wrinkled surface patterns. For this purpose, the photosensitive reaction mixtures were deposited via spin-coating over functionalized glasses. Following vacuum, UV, or either plasma treatments, it is possible to spontaneously form wrinkles, which could increase cell adherence. The pH responsiveness of the material was evaluated, observing an abrupt variation in the film thickness as a function of the environmental pH. Moreover, the presence of the carboxylic acid functional groups at the interface was evidenced by analyzing the adsorption/desorption capacity using methylene blue as a cationic dye model. The results demonstrated that increasing the acrylic acid in the microwrinkled hydrogel effectively improved the adsorption and release capacity and the ability of the carboxylic groups to establish ionic interactions with methylene blue. Finally, the role of the acrylic acid groups and the surface topography (smooth or wrinkled) on the final antibacterial properties were investigated, demonstrating their efficacy against both gram-positive and gram-negative bacteria model strains (*E. coli* and *S. Aureus*). According to our findings, microwrinkled hydrogels presented excellent antibacterial properties improving the results obtained for planar (smooth) hydrogels.

## 1. Introduction

Smart polymers are defined as materials that can actively alter their properties (such as volume or shape) in response to external changes [1]. For example, pH-sensitive materials can pass from open fully solvated coils to desolvated globular conformations over a small range of pH. These compounds generally contain some acid groups (–COOH, –SO_3_H) or basic groups (–NH_2_) in their chain. They have been utilized in different medical applications, mainly as drug vehicles or for gene and protein delivery applications [2], pH-sensitive materials for monitoring tissue acidosis [3], adsorbent materials for the remotion of cationic and anionic dyes [4], surface modification for contact lens [5], wound healing hydrogels with antibacterial activity [6,7,8], or as a pH-sensitive indicator to detect bacterial growth [9].

Polyelectrolytes are a particular type of smart polymers whose repeating units bear electrolyte groups, such as –COOH or –NH_2_ groups. Generally, the weak polyelectrolyte has an average dissociation constant (pKa or pKb) in the range of ~2 to ~10, which means that it would be partially dissociated at intermediate pH. Poly(acrylic acid) (PAAc) is a type of weak reversible polyelectrolyte that could accept or donate protons in response to pH variations. Interestingly, PAAc is widely used in applications for biomedical engineering, drug release, agriculture, and environmental protection [10]. Several methods have tested its pH-sensitiveness in the literature; for example, Sun et al. [11] synthesize microgels of poly(acrylic acid-*co*-vinylamine) (P(AAc-*co*-VAm)) for drug release applications. They studied the microgels’ zeta potential and swelling ratio, demonstrating that using P(AAc-*co*-VAm) makes it possible to generate a sustainable drug release system.

Similarly, Roghani-Mamaqani et al. [12] synthesized cellulose nanocrystals (CNCs)-grafted block copolymers of acrylic acid, N-isopropylacrylamide, and poly(N-isopropylacrylamide) (PNIPAAm). Dynamic light scattering (DLS) was used to measure the compounds’ hydrodynamic radius to find their lower critical solution temperature (LCST) and pH responsiveness. Interestingly, the block copolymers reversibly form a core-shell structure with PNIPAAm as core and PAAc as shell above LCST at higher pH values. Another example of PAAc used as a pH-responsive carrier for drug release applications was reported by Gupta and Purwar [13], who prepared hydrogel nanofibrous mats via electrospinning methods using PAAc and polyethylene glycol (PEG) as crosslinker. The nanofibrous mats were loaded with amoxicillin for antimicrobial activity. The results demonstrate that drug-loaded mats form a clear inhibition zone with gram-positive and gram-negative bacteria.

In some applications related to adsorption mechanisms, drug release, tissue engineering, soft machines, or soft robotics, the material should present on-demand switching from strong to weak surface adhesion [14]. Indeed, Yang et al. [15] reported an approach for switchable adhesion between hydrogels (P(PAAc-*co*-AAm) based on a mechanical wrinkling process. Thus, the formation and characterization of wrinkles are tailored by regulating the prestretch of adherend and optimizing adhesive ingredients synergistically. Furthermore, the topographic characteristics of the wrinkled patterns of different scales of length and amplitude in polymeric materials have been used to selectively control the adherence and proliferation of various cells [16]. These micro-nano structures allow the development of surfaces with effective antibacterial activity by direct contact (bactericidal) or preventing bacterial fouling at the surface (antifouling) [17]. Our research group worked with hydrogels with wrinkled surface patterns, based on the monomers 2-hydroxyethyl methacrylate (HEMA) and 2,2,2-trifluoroethyl methacrylate (TFMA), with poly(ethylene glycol) diacrylate (PEGDA_575_) as a crosslinking agent [18]. These studies demonstrated that the wrinkled patterns based on hydrogels with an amphiphilic balance in their components could act as selective membranes for cell growth and have properties such as biocompatibility and antibacterial activity.

These characteristics are essential because the transfer of microorganisms can quickly contaminate sterile polymeric materials in the medical area. By regulating the surface’s microstructure and adherence strength via pH-responsive changes, it is possible to fabricate switchable surfaces suitable for variable applications, like tissue engineering and drug/contaminant release/adsorption. Moreover, it has been demonstrated that PAAc presents antimicrobial properties against some antibiotic-resistant bacterial strains, often related to nosocomial infections [19,20]. Therefore, innovative material design approaches to achieve biologically active surfaces that prevent bacteria colonization are valuable in several biomedical-related fields [21].

This work aims to fabricate stable hydrogel films with reversible pH responsiveness that could spontaneously form wrinkled surface patterns and present antimicrobial properties against both gram-positive and gram-negative bacteria strains. The films comprise both AAc and PEGDA_575_ as monomer and crosslinking agent, respectively. Incorporating AAc in the reaction mixture affects the material’s swelling degree and the micelle’s dimensions, thus inducing changes in their hydrophilic nature and mechanical properties. More importantly, as will be discussed, these functional and wrinkled hydrogel films present important novel features compared to previously reported wrinkled surfaces. First of all, the presence of carboxylic acid groups will enable us to carry out chemical modification reactions. In this sense, the adsorption/desorption tests using methylene blue (MB) as a cationic dye will be described demonstrating that the wrinkled micropattern presence improves the material’s capacity to adsorb or release the MB. Moreover, the antibacterial performance of these materials, as evidenced by the LIVE/DEAD assays, was, on the one hand, significantly improved in the case of wrinkled surfaces in comparison to planar hydrogels. On the other hand, these materials are effective against both gram-positive (*Staphylococcus aureus)* and gram-negative (*E. coli*) bacteria.

## 2. Materials, Equipment, and Methods

### 2.1. Materials

The hydrogel composites were synthesized using different mole ratios of acrylic acid (AAc, 99.0%) and poly(ethylene glycol) diacrylate (PEGDA), with an average molecular weight of 575 g/mol. From now on, the composites *net-poly*(AAc-*co*-PEGDA_575_) will be designated according to their relative composition between AAc and PEGDA_575_ (0:1, 1:1, 5:1, 10:1, and 15:1). 2-hydroxy-4′-(2-hydroxyethoxy)-2-methylpropiophenone (Irgacure 2959, 98.0%) was used as a photo-initiator and methylene blue as a dye. All these reactives were acquired from Sigma–Aldrich (St. Louise, MO, USA) and were utilized as received without further purification.

3-hydroxytyraminium chloride (dopamine) was used to generate self-assembled monolayers (SAMs) on the glass substrates with the finality of favoring the adhesion with the hydrogel film. These reactives were acquired from Merck (Darmstadt, Germany). The tris(hydroxymethyl) aminomethane and acid chloride were purchased from Winkler Ltda. (Santiago, Chile). Round glass coverslips (nominal thickness: 0.13–0.16 mm) from Ted Pella, Inc., (Redding, CA, USA) were employed to deposit the hydrogel films via spin-coating. Quartz crystals (6 MHz, crystal area ø 6 mm) were acquired from Ted Pella Inc. (Redding, CA, USA).

### 2.2. Equipment

^1^H-NMR spectra were recorded on a Bruker 400 Ultrashield^TM^ spectrometer (Bruker Corp.,Billerica, MA, USA). The compounds were dissolved in dimethyl sulfoxide (DMSO-d_6_) using TMS (tetramethylsilane) as an internal standard at 60 °C.

A spin coater model KW-4A from Chemat Scientific (Northridge, CA, USA), coupled with a Rocker Chemker 410 oil-free vacuum pump from Rocker Scientific Co. (New Taipei, Taiwan), was used to deposit the reaction mixture over the pre-treated round glass coverslips. Photopolymerizations were carried out using a UV lamp (9 W) with an emission peak centered at λ = 365 nm from Vilber Lourmat Inc. (Marne-la-Vallée, France).

The polymeric films’ chemical composition and depth profiles were determined using a confocal Raman model CRM-Alpha 300 RA (WITec, Ulm, Germany) equipped with Nd:YAG dye laser (maximum power output of 50 mW at 532 nm). The Raman spectra were taken point by point with a resolution step of 100 nm. Cross-section images were acquired using this methodology. Attenuated total reflection-Fourier transform infrared (ATR-FTIR) was used to determine the chemical composition of the compounds in a Nicolet IS 5 FT-IR (Thermo Scientific, Waltham, MA, USA).

The hydrogels’ thermal stability was acquired by thermogravimetric analysis TGA-Q500 system (TA Instruments, New Castle, DE, USA). The data were obtained under a nitrogen atmosphere with a heating rate of 10 °C min^−1^, from ambient temperature to 650 °C. Additionally, enthalpy changes were analyzed with a differential scanning calorimeter, DSC-7 (Perkin Elmer Inc., Wellesley, MA, USA), under a nitrogen atmosphere at a heating rate of 10 °C/min. The glass transition (Tg) was obtained from the second heating cycle. 

Bresser Trino Researcher II (4–100×) trinocular microscope from Bresser GmbH (Rhede, Germany), coupled with a 5 Mp CCD color camera (Bresser GmbH, Rhede, Germany), was used as the first approach for visualizing the topography of hydrogel wrinkled surface patterns. Additionally, sample topographies were obtained at room temperature using an atomic force microscope (AFM) model NaioAFM, from Nanosurf AG (Liestal, Switzerland) in intermittent contact mode at different scan ranges (25 × 25 μm^2^ and 50 × 50 μm^2^). Images were treated using the offline software Gwyddion [22,23]. 

The particle size (hydrodynamic radius) at different concentrations and pH of the compounds, namely *net-poly*(AAc-*co*-PEGDA_575_), were obtained by DLS using a Zetasizer, model NanoS90 (Malvern Instruments Ltd., Malvern, UK), equipped with a 4 mW He-Ne laser (λ = 633 nm) at an angle of 90°. AR4 Abbe-refractometer from A. KRÜSS Optronic (Hamburg, Germany) was used to measure hydrogels’ refractive index. The contact angle using a ThetaLite optical tensiometer from Attension-Beloin Scientific (Gothenburg, Sweden) was measured. Experimentally, a droplet of 4 μL was gently deposited on the top of the films. This measurement was performed in three to five different film sectors to ensure reproducibility; three samples were also analyzed in each case. 

A multi-angle laser ellipsometer model SE400Adv, from SENTECH Instrument GmbH (Berlin, Germany), measured film thickness under different pHs (3.3 and 6.5). A Quartz Crystal Microbalance (QCM) was utilized to check the film’s thickness changes when the copolymer was exposed to different pHs. The QCM corresponds to a high-resolution thickness monitor system MTM-20 (Cressington Scientific Instruments Ltd., Watford, UK) designed for a sputter coater system. A Plasma Prep III system from SPI (Structure Probe Inc., West Chester, PA, USA) coupled to a process plasma controller and a vacuum pump model DUO 3 from Pfeiffer Vacuum GmbH (Asslar, Germany) was used for argon plasma exposure. Additionally, a V-730 UV-Vis spectrophotometer from Jasco Inc. (Easton, MD, USA) was used to evaluate the dye adsorption/desorption process.

For biological evaluation, *Staphylococcus aureus* (25293) and *Escherichia coli* (10536) bacteria strains were purchased from American Type Culture Collection (ATCC, Manassas, VA, USA). Luria-Bertani (LB) broth media (12780052) was purchased from ThermoFisher (Waltham, MA, USA). A spectrophotometer Specord 205 (Analytik Jena, Jena, Germany) was used to measure the optical density of bacteria culture. An inverted fluorescence microscope IX51 from Olympus Co. (Tokyo, Japan) was used to evaluate bacteria viability over wrinkled films.

### 2.3. Methods

#### 2.3.1. Step 1. Substrate Functionalization

First, the substrates were washed with a soap solution and rinsed with distilled water and then acetone and isopropyl alcohol to remove grease traces. Then, polydopamine was deposited in the glass slides’ surface according to the protocol reported by Nirasay et al. [24]. In this procedure, the glass was immersed in a 2 g/L dopamine solution in 10 mM Tris-HCl phosphate buffer (pH 8.5) and gently mixed on an orbital shaker for 3 h at room temperature to generate a SAM of polydopamine on the glass surface. The protocol followed in this case was not the same as Nirasay et al.; therefore, the film thickness (monitored via ellipsometry) resulted in a mean measured thickness of ~10 nm, slightly different than that obtained by the authors. Afterward, these substrates were rinsed with abundant deionized water, dried with ultra-pure nitrogen gas, and stored for further use.

#### 2.3.2. Step 2. Preparation of the Photosensitive Reaction Mixture

All the hydrogels were prepared in transparent flasks (with a septum) covered from light. These reaction mixtures were deposited over polydopamine-treated glasses. The amount of reactive used for the synthesis is shown in Table 1. The proposed chemical structure is depicted in Appendix A.

#### 2.3.3. Step 3. Deposition and Irradiation

The reaction mixture was deposited using the spin-coating technique to form homogenous hydrogel thin films. These homogeneous layers were obtained by applying subsequent velocity steps from 500 rpm, 1000 rpm, and 1500 rpm for 18 s. Nonuplets of each sample were deposited to ensure reproducibility and obtain essential data for the analysis, thus achieving statistically reliable results.

#### 2.3.4. Step 4. Deswelling, UV Light Irradiation, and Plasma Exposure Processes

Afterward, the hydrogel films were exposed to UV light irradiation (λ = 365 nm) for 5 min. Later, the samples were exposed to rough vacuum (1 × 10^−3^ torr) for 3 h to extract the material’s remnant solvent. This deswelling process generates two different zones on the material, a top rigid layer of more deswelled material and a soft hydrated foundation. The samples were then exposed to argon plasma (5–7 s), which serves as an external stimulus to spontaneously generate wrinkled patterns on the film’s surface due to the mechanical interaction of the deswelled layer and the hydrogel’s soft foundation. This process also probably oxidizes and polymerizes the top layer of the film due to the generation of free radicals on the surface via frontal vitrification [25]. Finally, the sample is exposed to UV light for 30 min to fully polymerize the film and fix the hydrogel surface’s wrinkled pattern (Figure 1).

To generate smooth (flat) films, the reaction mixture previously deposited by spin-coating was exposed to UV-light for 35 min to generate a total polymerization of the material. Afterward, the films were exposed to vacuum for 2 days and finally irradiated with argon plasma.

### 2.4. Evaluation Tests

#### 2.4.1. Dye Adsorption/Desorption Tests

Adsorption/desorption experiments were carried out using two hydrogel films (with and without wrinkles). Following a typical experimental procedure [26], a dye solution of MB with a concentration of 5 × 10^−3^ mM was prepared at pH 7 to test its adsorption/desorption kinetics. Subsequently, on a glass coverslip (1 cm^2^), with the hydrogel film on the surface, the MB dye mixture was added upon stirring at a speed of 400 rpm for 300 min. Approximately 600 µL of the dye solution was extracted at different times. The decrease in absorbance of the typical MB peaks was monitored using a UV-Vis spectrophotometer. The dye concentration was calculated at each time using the Beer–Lambert law [27]. Finally, desorption tests were carried out by immersing each MB-loaded hydrogel in an aqueous HCl solution at pH 2 for 60 min. All tests were carried out at room temperature and atmospheric pressure.

#### 2.4.2. Antibacterial Evaluation

Samples were first sterilized in a 12-well plate, using two initial washes with 1 mL of phosphate-buffered saline (PBS). Subsequently, surfaces were exposed to 40 min of UV light before a final wash with PBS. For bacteria inoculum, 1:10 dilutions of previously obtained 0.8 O.D. bacterial suspensions (*Staphylococcus aureus* and *Escherichia coli*) stored at 4 °C were prepared in LB media and left at 37 °C under constant shaking of 125 rpm for 18–24 h. Next, it was verified that the optical density of bacteria cultures was 0.8 at 600 nm using a spectrophotometer previous to cell seeding over films. Then, 1 mL of this inoculum was added to each sample and incubated for 1 h at 37 °C. Once the inoculum was removed, bacteria adhered to the surfaces to grow in PBS for 48 h. LIVE/DEAD™ BacLight™ (ThermoFisher Scientific) kit test was used to evaluate biofilm formation and viability. Staining was performed for 15 m in dark conditions at room temperature, followed by rinsing twice with PBS. Samples were photographed using a fluorescence microscope (Olympus IX 51). This essay analyzes the membrane integrity of the cell, staining green (SYTO^®^ 9) if they are intact or red (propidium iodide) if the membrane is compromised. Green fluorescent bacteria were photographed using FITC filter (λ_ex_/λ_em_ = 490/525 nm), and red fluorescent-labeled cells were observed with a TRICT filter (λ_ex_/λ_em_ = 550/600 nm). Images were acquired by triplicate using 200× magnification, and bacterial coverage was calculated using the software ImageJ.

## 3. Results and Discussion

### 3.1. Preparation of the Wrinkled Functional Surfaces

The preparation of the stimuli-responsive wrinkles was carried out following the scheme depicted in Figure 1. The hydrogel films were firstly formed via spin-coating technique over polydopamine-functionalized round glass coverslips. The films were then exposed to UV, vacuum, and argon plasma to trigger the wrinkled patterns on the hydrogel surface. Finally, the films were exposed again to UV irradiation.

It has been reported that the plasma treatment induces surface oxidation and an increase in the polymerization degree of the material [28]. To confirm this affirmation, two representative samples (5:1 and 15:1) were selected to perform ATR-FTIR. These results show an increase in the carbonyl group vibration at 1722 cm^−1^ after argon plasma irradiation. The percentage increment in the peak integrated area was 64.9% and 99.4% for the samples 5:1 and 15:1, respectively. In Appendix A, is possible to observe the spectra for both mentioned samples, normalized to the highest intensity peak at 903 cm^−1^, attributed to C–O–C symmetric stretching band. 

As expected, both FT-IR Spectra showed an increase in three signals (2866 cm^−1^, 1728 cm^−1,^ and 1093 cm^−1^). The signal intensity increase confirmed that the oxygen-containing groups were incorporated onto the surface via the plasma treatment. Similar behavior was observed on the signals at 2944 cm^−1^ and 2866 cm^−1^, related to the antisymmetric/symmetric vibration of –CH_3_ and –CH_2_– stretching groups. At ~1728 cm^−1^ and ~1093 cm^−1^, the C=O vibration and C–O–C asymmetric stretching band were shown, respectively.

### 3.2. Characterization of the Hydrogels before Film Deposition

^1^H-NMR, DLS, and thermal studies (TGA and DSC analyses) were performed using the wrinkled hydrogel films deposited over a glass substrate (non-functionalized). The solid film was detached from the substrate by immersing them in nanopure water in all the cases. Afterward, the films were dried under vacuum at 60 °C. Then, the samples were grounded, and the obtained powder (dry hydrogel) was utilized to perform chemical and physical characterization.

Information about the monomer and crosslinking agent conversion and therefore the wrinkled hydrogels’ chemical composition was obtained via ^1^H-NMR and Raman confocal spectroscopy. To carry out an exhaustive analysis of the signals observed by ^1^H-NMR, the data were compared to the polymeric form of the two isolate ingredients (polyAAc and polyPEGDA_575_). The polyPEGDA_575_ was synthesized by following the same copolymerization methodology and was coined as 0:1. In the case of polyAAc, it was synthesized using the procedure proposed by Rafi Shaik et al. [29], which was corroborated by ATR-FTIR (Appendix A). By analyzing the ^1^H-NMR signals and the area under each peak (Appendix A), it is possible to demonstrate the formation of a crosslinked structure and determine the monomer and crosslinking agent conversion [18]. Detailed identification of ^1^H-NMR signals is performed in the Appendix A. To determine the concentration of AAc that reacted (Table 2), two signals from the NMR spectrum were analyzed: from the bare monomer (AAc) at 6.11 and 6.04 ppm (CH_2_=CH–C(O)–OH) and from their polymeric backbone chain at 4.10 and 4.08 ppm (–CH(COOH)–CH_2_)_n_–). The second signal is related to the total monomer present in the sample, both from the monomer that reacts and the one that does not. Thus, comparing these values, it is possible to obtain the amount of monomer that effectively reacts. Thus, the conversion degree can be calculated by dividing the integral of the respective polymer peak by the sum of integral monomer peak+integral polymer peak. Additionally, the signals selected to determine the concentration of PEGDA_575_ that reacted were two: from the bare monomer (PEGDA_575_) at 6.21 and 6.16 ppm (CH_2_=CH–C(O)–OR) and their polymeric chain at 3.68 and 3.63 ppm (–CH_2_–CH_2_–C(O)–O–CH_2_–).

The results of polymeric percentage conversion and the expected composition are shown in Table 2. Compared to the expected feed composition, the copolymer formed presents a slightly lower concentration of PEGDA_575_ than expected according to the amount added in the reacting mixture. All the reaction samples present a real composition remarkably similar to the expected one according to the amounts of reactives introduced in the mixture. Similar results have also been found in previous studies performed by our research group [14,18].

In addition to the ^1^H-NMR, the chemical analysis of the wrinkled hydrogels synthesized using different feed mole ratios was carried out via Raman spectroscopy (Figure 2). These results show the characteristic bands for this kind of hydrogels, such as the antisymmetric/symmetric vibration of –CH_3_ and –CH_2_– groups, located at 2941 cm^−1^ and 2880 cm^−1^, respectively. In addition, it is possible to detect that the peak at 1723 cm^−1^ corresponds to the carbonyl group from PEGDA_575_ or AAc (>C=O or –COOH) and has a significant intensity increase when the inclusion of AAc in the mixture occurs, which is expectable. A similar situation occurs with the band frequency at 1634 cm^−1^ and 1601 cm^−1^, corresponding to the C=C stretching mode, which could also be utilized to measure the degree of crosslinking of the polymer meshwork. Thus, when increasing the mole ratio (from 0:1 to 15:1), the copolymer presents more free double bonds due to PEGDA_575_ as a crosslinking agent. Similar behavior is observed for the bands in the range 1460–1480 cm^−1^, associated with the antisymmetric –CH_3_ bending and –CH_2_– scissoring. 

An intense signal is observed near 1571 cm^−1^ due to the stretching vibration of the conjugated system of –C=C– and >C=O groups. This band suddenly appears in the samples 15:1 and 10:1, behavior related to the AAc concentration utilized in the reaction mixture (copolymer). On the other hand, the characteristic Raman signals from carboxylic groups can be observed at 1409 cm^−1^, which corresponds to the interaction of oxygen-hydrogen in-plane bending (C–OH in-plane bend). From 1301 cm^−1^ to 1280 cm^−1,^ two coupled signals could be observed: skeletal vibration –(CH_2_)_n_– in-phase twist and C–O stretching. These bands’ intensities also increased according to carboxylic acid present in the hydrogels’ surfaces. These variations could also be related to the polymerization degree of each composite. Finally, four signals with strong-medium intensities are shown at 1128 cm^−1^, 1035 cm^−1^, 852 cm^−1^, and 815 cm^−1^, which should be related to C–C skeletal stretching.

In Figure 3 are depicted the TGA and the DSC traces for the different hydrogels, and in Table 3 are summarized the most relevant results obtained from the TGA and DSC curves. From TGA data, it is possible to determine the weight loss at different temperatures (260 °C, 340 °C, and 460 °C). At 260 °C is shown a slight weight loss increase from 0.8% to 14.0% and at 340 °C from 6.0% to 27.2% when the amount of AAc is increased in the mixture from 0 to 15 mole (0:1 to 15:1). Similarly, it is possible to observe a residual mass increase from 3.3% to 9.1% at 460 °C. These results show that the thermal stability of the compound slightly decreases with the AAc inclusion, leaving a more significant residual mass at high temperatures. Jeong et al. [30] obtained similar results using *poly*AAc hydrogels, which contain different metronidazole concentrations. According to the obtained thermograms, three main steps of thermal decomposition are shown: the first one is related to the evaporation of adsorbed water (weight loss about 0.2–2.7%); the second could be attributed to the destruction of *poly*AAc lateral groups, leading to decarboxylation or anhydride formation (weight loss of ~2.6–11.9% between 200 °C and 300 °C); and the third step is associated to main chain scission and composite depolymerization above 300 °C. 

By analyzing the DSC curves, the glass transition temperature (Tg) was determined as the temperature at the mid-point of the endothermic rise, measured from the post-transition baselines’ extension [31]. Thus, the five hydrogels’ DSC curves possess only one Tg each, which increased from −29.2 °C to 22.7 °C with the increasing inclusion of AAc in the mixture (from 0 to 15 mole). These Tg values are associated with the concentration of monomer or crosslinking agent used for synthesizing the hydrogel. According to the literature, the Tg of *poly*AAc is 123 °C [32,33], and for *poly*PEGDA_575,_ the Tg was found from −40 to −30 °C [34]. It is possible to conclude that the Tg values obtained for the composites are in concordance with each hydrogel’s compositional ratio, showing an increase when AAc is included in the mixture, as was expectable.

### 3.3. Wrinkling on Solid Supports to Produce Stimuli-Responsive Microstructured Hydrogels

The successive vacuum and UV irradiation steps lead to wrinkled hydrogel films with variable wrinkle characteristics (wavelength and amplitude). In addition to the preliminary visualization of the wrinkle formation by optical microscopy (Appendix A), AFM microscopy was used to obtain more accurate information. Figure 4a shows a set of AFM images for the samples at different mole ratios (from 0:1 to 15:1). Additionally, the top right corner inset represents the two-dimensional Fast Fourier Transform (2D-FFT) of the respective AFM micrographs. Information about the wavelength, amplitude, roughness, area increase percentage, and aspect ratio (wrinkle height/width) can be obtained by analyzing these images. Finally, the samples’ thicknesses can also be determined by measuring the border’s depth profile (see Figure 4b). For all the examples, the film thicknesses were found in the range of 1.5–1.7 μm, thus indicating that the spin-coating technique generates homogenous films with similar thicknesses independently of the sample composition used in each case, as expected. These results were later corroborated via ellipsometric techniques.

The distribution of the wrinkled patterns is not significantly affected by the variation of hydrogel composition; in all cases, the distribution is homogenous in all directions, forming herringbone-like structures [35]. The 2D-FFT results allow us to demonstrate this affirmation due to concentrical rings with different intensities, which is a typical pattern characteristic of homogeneous wrinkled distributions that do not have any preferred ordering direction [36,37,38]. This kind of homogenous herringbone-like structure is typical of free equi-biaxial contractions [39,40], which, in this case, was produced by a mechanical contraction as a result of the deswelling process when the sample is dried under vacuum.

As it is possible to observe from the data, the width (wavelength) of the wrinkles increases dramatically when the mole ratio concentration changes from 0:1 to 1:1 (Figure 5a); this could be related to the inclusion of AAc in the mixture, which could alter the mechanical properties of the material, thus affecting the wrinkling process mediated by surface instabilities. Then, for the samples 5:1, 10:1, and 15:1, is possible to detect a slight decrease in wrinkle width with the inclusion of AAc in the mixture. Interestingly, the wrinkle height (amplitude) and wrinkle width (wavelength) possess a similar behavior for these three samples (Figure 5a). This tendency is also reflected in the wrinkles’ aspect ratio (amplitude/wavelength), whose values are between 0.26 ± 0.13 and 0.51 ± 0.21 (Figure 5c). Aspects ratios between those ranges indicate that these samples’ wrinkled patterns are considered ripples or small ridge patterns [41]. A similar situation occurs for the roughness and area increase percentage measured for each sample; an increase from sample 0:1 to 1:1 is detected, probably due to the AAc insertion, and then a slight decrease is observed for the samples 5:1, 10:1, and 15:1. The area increase percentage was obtained from the AFM micrographies via the determination of the real surface area and the projected area of the analyzed sector. 

As demonstrated by several research groups [42,43,44] and also reported by our group [18,45,46], the variation of microstructural patterns, particularly wrinkled patterns, could affect material biocompatibility and cell adherence due to some microstructural cues present on the surface of the material. A priori, the AAc-based hydrogel films fabricated in this work are expected to improve their compatibility with biological tissues by altering their wrinkled microstructural patterns. 

### 3.4. Evaluation of the pH Response of the Wrinkled Surfaces

Several complementary techniques were employed to characterize the wrinkled hydrogel surfaces’ pH sensitiveness, i.e., DLS, water contact angle, QCM (Quart crystal microbalance), and ellipsometry measurements.

DLS was carried out for aqueous solutions of the copolymers. The hydrogels in the buffer (3 mg of the copolymer in 5 mL of pH) were agitated in small flasks for 72 h and sonicated to ensure a complete hydrogel dissociation. Hydrodynamic radius, using distilled water and pHs 3.3 and 6.5 buffers, were obtained through this method. The introduction of -COOH groups in the hydrogel alters the interaction with water at a determined pH. Thus, at pH lower than 4.5, most -COOH groups from the AAc are protonated, while at higher pHs, they are ionized, leading to broken hydrogen bonds and swollen microgels [47]. Figure 6a shows the microparticles’ hydrodynamic radius as a function of pH. In general, as is possible to observe, at pH 1.0, the hydrodynamic radius is lower than at pH 9.0. This behavior is related to the stretching of the molecules and their hydration. At low pH, *poly*AAc adopts a compact (but not fully collapsed) globular conformation. However, ionization occurs when the pH increases above its pKa, and the polymers expand into a fully solvated, open-coil conformation [48]. With this, it is possible to observe that the hydrodynamic radius increases with AAc concentration in the composite. Moreover, the difference between the hydrodynamic radius at pHs 3.3 and 6.5 becomes larger when AAc increases. Interestingly, at concentrations of 10:1 and 15:1, the difference in the hydrodynamic radius for the different pH stops growing, indicating that a plateau is reached over 10:1 concentration of AAc. Additionally, from the DLS-Titration curve, it is possible to conclude an increase in hydrodynamic radius upon an increase in pH, behavior related to a possible molecular association started at pH ≥ pKa (located at 4.5). These results show the familiar expansion of the polymeric chain upon the pKa; similar behavior was studied by Swift et al. [49] in the polymers *poly*AAc and *poly*(AAc-*co*-ACE) (ACE, Acenaphthylene). Whereas the DLS analysis provides information about the capability of the size and polydispersity of the swollen hydrogels in a dispersed state, with the finality of characterizing the hydrophilic or hydrophobic nature of the materials, static contact angles were measured using the sessile drop methodology.

As depicted in Figure 6c, is possible to conclude that copolymers with the highest amount of acrylic acid in their main chain show the lowest contact angle due to the incorporation of polar groups (–COOH), which increase the hydrophilicity of the hydrogels. Additionally, the pH-sensitive capacity of copolymers was studied through contact angle measurement at different pH (1.0 to 9.0). Thus, at increasing the pH values (≥pKa [50,51]), a slight decrease in the contact angle for all the samples that contain acrylic acid segments (1:1 to 15:1) is observed. These low values are related to –COOH functional groups in the polymer chains, predominantly undissociated (at pH 3, these groups’ dissociation degree (α) is equal to 0.03). At pH 4.5, the number of –COOH groups is the same as –COO− (*α* = 0.5). As the pH rises above 4.5, polyelectrolyte dissociation increases rapidly; at pH 6, it equals 0.97, and at pH 9, reaches values close to 1 [52,53,54].

Palacio-Cuesta et al. [55] studied the variation of the contact angle as a function of the wt% of copolymer *poly*(MMA-*co*-AA) in flat and wrinkled surfaces. Thus, the planar surface slightly decreases the contact angle in comparison with wrinkles surfaces.

According to the results obtained via DLS analysis, the samples 10:1 and 15:1 presented the larger changes with the pH. These two were selected to address the hydrogel films’ physical changes when exposed to different pH mediums using QCM and ellipsometric measurements (Figure 7).

A QCM consists of a thin quartz disc placed between a pair of gold electrodes. Due to quartz’s piezoelectricity, it is possible to excite a shear mechanical oscillation of the crystal by applying an AC voltage across the substrate. The sensor’s resonance frequency depends on the total oscillating mass, including any medium or material coupled to the substrate. The resonance frequency follows the Sauerbrey relation, which is inversely proportional to the film’s areal mass density (including the solvent within the film) [56]. Indeed, QCM measures changes in the material’s mass density, which could be related in some cases to thickness increase. In this case, the hydrogel thin films absorb or release solvent depending on the medium’s pH, which variates the sample’s mass density and possibly the film thickness. Similar studies were performed by Borisov et al., which utilized the QCM to analyze the effect of pH and electrical stimuli on brushed grafted polymers [57].

In our case, two buffers based on citric acid and sodium hydroxide with pH 3.3 and 6.5 were used to test the copolymer’s pH response. A small drop of the reaction mixture dissolved in methanol (200 μL of the mixture in 4 mL of dissolvent) was deposited on the quartz sensor and irradiated with UV light for their polymerization. The drop was allowed to evaporate at ambient conditions for few minutes, and the frequency measurement was carried out. Then, the process was repeated using a different buffer. Figure 7a shows the time dependence vs. thickness percentage variation of the samples (10:1 and 15:1) with three pH cycles. Thus, when the pH is increased from 3.3. to 6.5, the carboxylic groups become ionized, the AAc chains are expected to stretch, and the brush becomes thicker and increasingly hydrated, thus increasing its mass density and probably increasing its thickness. The QCM response was reversible and fast (less than five minutes) when the pH was decreased from 6.5 to 3.3, reaching a similar value as the measured initially. The percentage thickness difference is almost the same for the two concentrations of AAc analyzed (10:1 and 15:1), similar to the hydrodynamic radius results obtained from the DLS analysis. The compounds that present lower concentrations of AAc, like 1:1 or 5:1, do not show any significant thickness variation.

Ellipsometry measurements were also performed on the thin, solid hydrogel films to detect physical changes and corroborate the QCM results. Biesalski et al. [58] described a weak polyacid brush’s synthesis and swelling behavior attached to a solid surface. Thus, pH values (with and without monovalent salts) affect the layers’ thickness, monitored by multiple-angle null-ellipsometry. In our case, ellipsometry measurements were performed to corroborate the results obtained by QCM. Thus, the reaction mixture composed of AAc and PEGDA_575_ was dissolved in methanol (200 μL in 800 μL of dissolvent) and deposited via spin-coating on SAMs-silicon wafer (SAM thickness was previously studied by this technique).

Afterward, the films were polymerized using UV light and immersed in water; then, the thickness was studied in dry and swelling states at different pHs. Figure 7b shows the swollen thickness of two different hydrogels (10:1 and 15:1) as a pH (6.5 and 3.3) function. By increasing the pH of the medium (from 3.3 to 6.5), the hydrogel film thickness shows significant conformational changes related to the number of dissociated carboxylic acid groups, i.e., a higher degree of dissociation implies a higher charge density, a higher osmotic pressure of the counterions, and therefore an increased thickness of the film. The thickness measurement was repeated cyclically in the same spot by varying the pH on the medium (seven pH cycles were performed for each sample), showing similar results in each cycle, i.e., a thickness increase when the pH increases from 3.3 to 6.5. Finally, and similarly to the results obtained from QCM and DLS data, the film thickness difference between pH 3.3 and pH 6.5 did not vary significantly between the two concentrations (15:1 and 10:1). This effect could probably be related to the molecular weight of the obtained compounds after deposition. According to Borisova et al. [57], the variation of micelle hydrodynamic radius depends on the compounds’ molecular weight and not on the compound’s chemical conformation. According to the NMR spectra, the nominal molecular formula weight of the repetitive units for each sample (10:1 and 15:1) are similar (~1220 g/mol and ~1510 g/mol, respectively); therefore, it is expectable that the film thickness variation between pH 3.3 and pH 6.5 is comparable. The film thickness variation could be an exciting capability for materials intended to be used in biomedical or tissue engineering due to its capacity to actively change its conformation under different pH mediums, useful as drug carriers or biosensors. Finally, for films with lower concentrations of AAc (1:1 and 5:1), the thickness percentage variation was small enough to be considered as measurement errors, so it was not included in the study. This effect could be related to the swelled state of the polymeric films. Small, conformational changes are easily detectable when the compound is in solution due to the molecules’ freedom degrees. In a swelled state, the molecule is restricted; thus, the conformational changes produced due to pH variations are difficult to detect.

Finally, the pH response has not only an effect on the film thickness but also on the wrinkle characteristics, i.e., wavelength and amplitude. As depicted in Figure 8, both amplitude and wavelength increased with the pH. The changes observed are reversible upon changing the environmental pH. Interestingly, this system based on acrylic acid complements previous works in which the use of diethylaminoethyl methacrylate (DEAEMA) (thermal-pH responsive) allowed the preparation of wrinkles with lower wavelength and amplitude upon increasing the pH [14]. It is interesting to note that, in those previous works, the wavelength of the wrinkles formed remains clearly above 5 μm, while with this current approach, the wavelength has been significantly reduced to the range of 1.2–1.7 μm.

Finally, the availability of the carboxylic acid groups to establish interactions with ionic molecules was investigated by the adsorption of methylene blue [26]. Hydrogel-based biopolymers are one of the materials most used as adsorbents due to their ability to absorb and retain large amounts of water in their structure. This behavior is mainly due to the presence of polar functional groups that strongly interact with water molecules, such as amino (–NH_2_), hydroxyl (–OH), and carboxyl (–COOH) groups [59]. These hydrogels’ acidic or basic groups can accept or release protons due to pH variations, generating charges on the polymeric structure. In carboxylic acid, at a pH above pKa, the –COOH groups dissociate to form carboxylate ions (–COO–), which strongly interact with cationic dyes, like MB [60]. Accordingly, methylene blue (MB) adsorption and desorption studies were carried out in smooth and wrinkled hydrogel samples to confirm the hydrogel’s response capacity at pH above and below pKa and the influence of wrinkled morphologies on adsorption. It is expected that samples with micro-patterns on their surface should present a higher surface area for contact with the medium, thus increasing its adsorption/desorption capacities. 

Accordingly, Figure 9a shows the results obtained by UV-Vis spectroscopy at pH 7. The changes in the concentration of MB during the removal process are monitored by following the most intense adsorption band (664 nm). It was observed that after 120 min of magnetic stirring, the adsorption rate remained constant, managing to remove 61.4%, 23.9%, 82.9%, and 42.3% in the samples 0:1 (wrinkled), 0:1 (smooth), 15:1 (wrinkled), and 15:1 (smooth), respectively. The results show that the incorporation of the AAc in the polymer produces a considerable increase in the adsorption capacity of MB, as was expectable. These results are also observable in Figure 9c, where the hydrogels with a 15:1 concentration present an intense pigmentation compared to the 0:1 concentration. Additionally, it is possible to observe that the presence of a microwrinkled pattern on the hydrogel films generates a significant increase in adsorption capacity of the material (an increase of 40.7% for sample 0:1 and 41.3% for sample 15:1), which is directly related to the rise in the contact area when forming micro-morphologies to surface level (Figure 5b). With pH values below the pKa, the adsorption percentage of the dye is controlled mainly by the –COOH groups of the adsorbent. Under this condition, the –COOH dissociate to form –COO–, where the negatively charged oxygen atom generates an electrostatic attraction with the positive charges of the =N^+^ fractions of the cationic dye [61].

The hydrogel samples loaded with MB were immersed in an acidified aqueous solution (pH 2) for 60 min to conduct the dye desorption tests. In these processes, a 25.8%, 4.5%, 56.2%, and 19.9% desorption was achieved for the samples 0:1 (wrinkled), 0:1 (smooth), 15:1 (wrinkled), and 15:1 (smooth), respectively (Figure 9b). Although both concentrations manage to desorb part of the adsorbed dye, the values for the 15:1 sample are higher than in hydrogels without acrylic acid (0:1). This behavior confirms the pH responsiveness of hydrogels with carboxylate functional groups, which lose their charge at pH below pKa. Therefore, electrostatic interactions between adsorbate and dye decrease, increasing the desorption percentage, as was expected.

### 3.5. Antibacterial Evaluation of the PAA Functionalized Hydrogel Wrinkled Films

Provided the capacity of the carboxylic acid groups to establish interactions, the capacity of these functional groups to act as antibacterial materials for biomedical purposes was explored. For this purpose, smooth and wrinkled films with increasing acrylic acid content (1:1, 5:1, and 15:1) were evaluated as antibacterial surfaces. In this evaluation, *Staphylococcus aureus* was chosen as a model due to their implication in nosocomial infections, especially with antibiotic-resistant strains. In addition, the *Escherichia coli* bacterial strain has also been evaluated as a gram-negative model. Thus, a solution with the bacteria was allowed to adhere to the wrinkled films and then proliferate for 48 h. After that, the films were removed from the incubator, and the viability of the bacterial cells at the film surface was evaluated using a LIVE/DEAD fluorometric staining.

Figure 10a shows live (green) and dead (red) *S. aureus* bacteria over films (wrinkled, smooth, and control (C)). The samples designated as control (C) correspond to a planar surface formed exclusively by PEGDA_575_, i.e., without acrylic acid. Over smooth control surface, viable bacteria colonies were observed. However, the presence of acrylic acid induces the apparition of red-colored bacteria, i.e., dead bacteria. Interestingly, when the acrylic acid content increases on smooth surfaces, the number of dead bacteria becomes higher, evidencing the antimicrobial behavior of this surface functionalization.

Moreover, even fewer isolated living bacteria were photographed over wrinkled samples, whereas a high amount of dead clusters were detected on the three different wrinkled surfaces. In all wrinkled samples, dead bacteria’s percentage of area occupied is significantly higher than occupied by live bacteria, demonstrating an enhancing action compared to their smooth counterparts (Figure 10b). This observation can be explained by considering that the microwrinkled surface structure favors the bacterial attachment to the surface. Once attached (most probably on the valleys of the wrinkles), they are exposed to a larger amount of antimicrobial acrylic acid groups.

Furthermore, the total *S. aureus* coverage of wrinkled films (Figure 10b), especially dead bacteria’s percentage of the occupied area, shows a descending trend as the proportion of AAc augments. This fact could be a consequence of the first initial cell adhesion steps, where wrinkled surfaces with a high proportion of AAc could prevent *S. aureus* colonization showing an additional antifouling effect. After that, bacteria proliferation over wrinkled films was prevented, as viability was compromised due to the AAc action in the bacteria lipid membrane and peptidoglycan layer.

In addition, a similar evaluation was carried out using *E. coli* as an experimental model. Figure 11 shows the antimicrobial activity for smooth and wrinkled samples, with a dramatic decrease of bacteria viability in all films containing acrylic acid. In this case, higher proportions of AAc provoked an increase of dead *E. coli* surface coverage (Figure 11b), showing a remarkable efficacy for 15:1 AAc wrinkled films. Therefore, the antifouling effect was not detected using *E. coli* as an experimental model, suggesting a different early interaction between AAc films and gram-negative bacteria. However, *E. coli* contact killing was also improved using wrinkled surfaces, demonstrating the potentiality of this strategy. Different polymeric supports have previously discussed this synergic behavior between surface composition and topography [16], showing results similar to the obtained but without the pH responsiveness capacity of the AAc-based wrinkled hydrogel film presented in this study. Finally, these results demonstrate an apparent antibacterial effect for all evaluated wrinkled samples, suggesting their application as coatings for clinical devices or biomedical applications.

## 4. Conclusions

Hydrogel composites were synthesized using different mole ratios of AAc and PEGDA_575_. ^1^H-NMR studies demonstrate that the procedure followed to synthesize the polymers results in almost the same composition as theoretically expected. Together with this, TGA and DSC were also carried out to investigate the samples’ thermal behavior. These results show that the weight loss increases with the amount of AAc included in the mixture. A similar situation occurs for the residual mass at 460 °C and the Tg values, increasing from −29.2 to 22.7 °C with an AAc increase, as expected. 

Once the compounds’ chemical and thermal characterization were finished, the samples were spin-coated spin over SAM-dopamine functionalized glasses to form thin films. Then, wrinkled patterns were formed spontaneously at the top of the films by exposing the samples to UV radiation, vacuum, and argon plasma. Raman spectroscopy was used to corroborate the deposited samples’ chemical composition, showing that characteristic bands of the AAc and PEGDA_575_ could be detected in the appropriate proportions for each compound, indicating that the sample deposited maintains the chemical structure of the synthesized composite. Using AFM, the wrinkled patterns’ dimension was characterized, showing a stabilization of the wrinkle wavelength and amplitude for the samples 5:1, 10:1, and 15:1.

Contact angle measurements were carried out to analyze the samples’ polar nature at different pHs. These results indicate that at larger pHs, the contact angle of the wrinkled surfaces increases for all the composites, including AAc, resulting in more hydrophobic samples when the pH increases over AAc pKa. This study aimed to fabricate hydrogel films with reversible pH responsiveness that spontaneously form wrinkled surface patterns; to this end, QCM and ellipsometry measurements were carried out on the thin films, resulting in notable thickness variations of the films depending on the pH medium. The hydrogels’ response to pH changes was fast and could be entirely rationalized based on the theory of conformational changes. The adsorption/desorption kinetics of the cationic dye (MB) at different pH allow to confirm the response capacity of hydrogels against pH and determine the influence of surface modifications on the interaction of external agents comparing the wrinkled and smooth hydrogel samples. An increase in the contact surface area produces a considerable increase in the adsorption of dyes, which is of great importance to address the current environmental problems associated with colored industrial waste.

Interestingly, the inclusion of AAc in the hydrogel mixture affects the dimensions of the wrinkled patterns, theoretically altering the material capacity to interact with biological matter and dyes. On the other hand, despite these physical changes, the pH responsiveness of the material (QCM and ellipsometry) or their polar nature (contact angle) does not seem affected, thus becoming a proper smart material for biomedical and dyes adsorption applications. Furthermore, an antibacterial effect has been detected for all AAc wrinkled films, showing an improved efficacy for *S. aureus and E. coli* strains compared to their smooth counterparts. In this scenario, AAc hydrogel coatings could efficiently avoid bacterial proliferation and prevent nosocomial infections in clinical scenarios. 

## Figures and Tables

**Figure 1 polymers-13-04262-f001:**
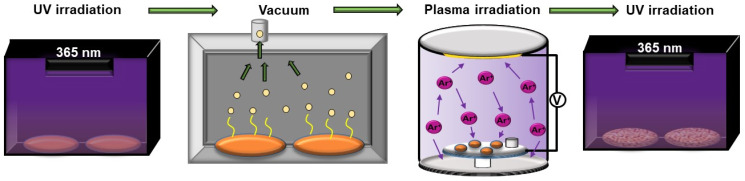
Schematic representation of the subsequent steps used to generate the wrinkle patterns.

**Figure 2 polymers-13-04262-f002:**
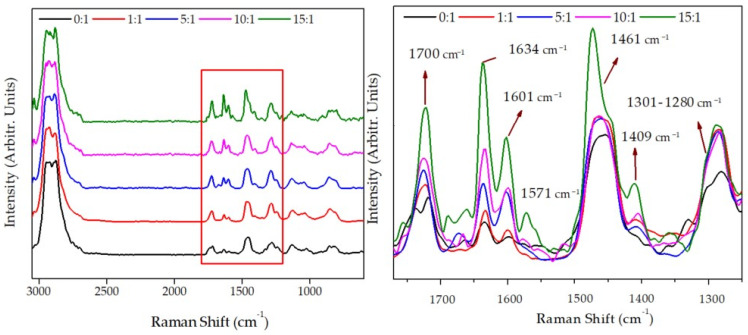
Raman spectra of *net-poly*(AAc-*co*-PEGDA_575_) at a different mole ratio, i.e., AAc:PEGDA_575_ of 0:1, 1:1, 5:1, 10:1, and 15:1.

**Figure 3 polymers-13-04262-f003:**
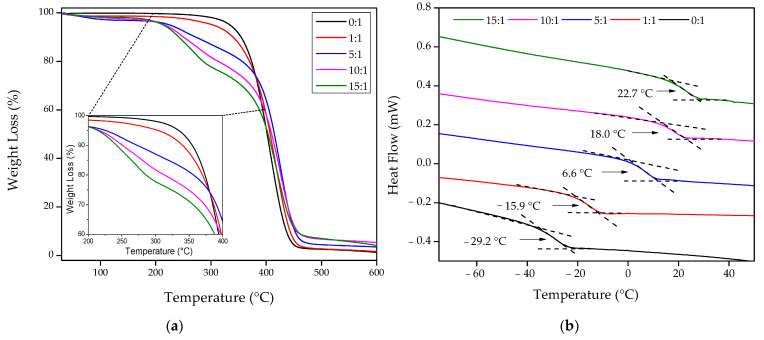
Thermal analysis of the wrinkled hydrogel surfaces with different chemical compositions: (**a**) TGA and (**b**) DSC.

**Figure 4 polymers-13-04262-f004:**
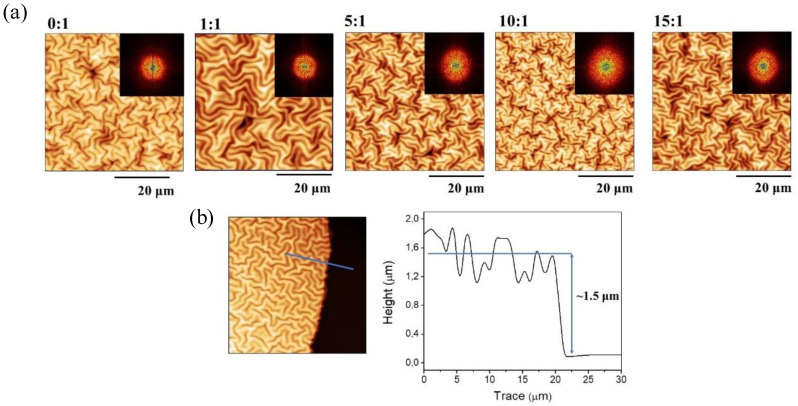
(**a**) AFM topography images of the wrinkled patterns and their corresponding 2D-FFT and (**b**) film thickness measurement using AFM images.

**Figure 5 polymers-13-04262-f005:**
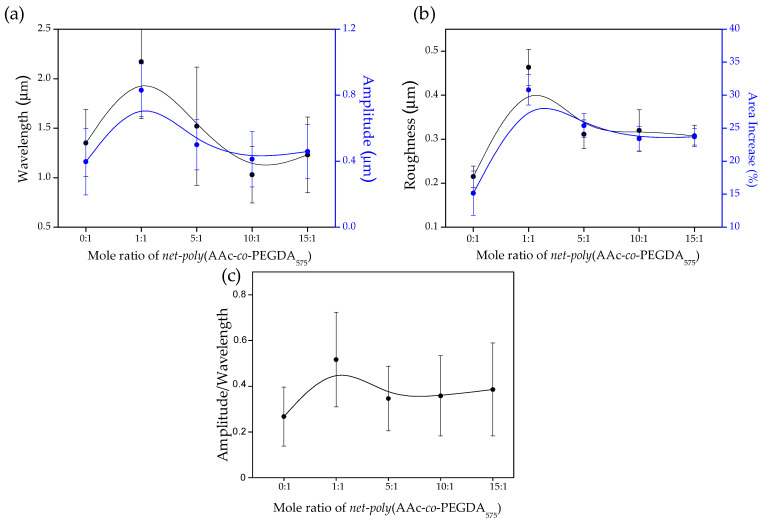
Morphological characteristics obtained from AFM analysis. (**a**) Wrinkle wavelength and amplitude, (**b**) roughness and area increase, and (**c**) aspect ratio of the hydrogel films of *net-poly*(AAc-*co*-PEGDA_575_) at a different mole ratio (0:1, 1:1, 5:1, 10:1, and 15:1).

**Figure 6 polymers-13-04262-f006:**
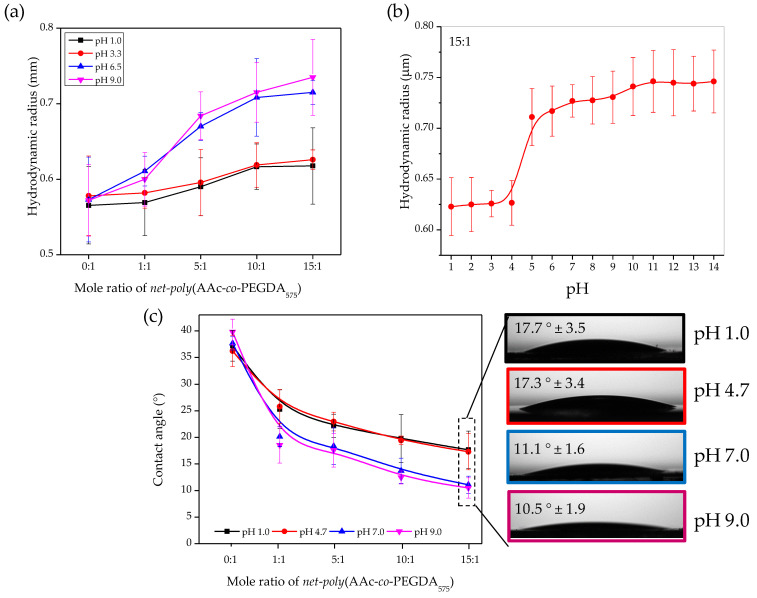
(**a**) Dependence of the hydrodynamic radius at four different pHs (1.0, 3.3, 6.5, and 9.0) for the samples *poly*(AAc-*co*-PEGDA_575_), (**b**) DLS-Titration curve of the for the sample 15:1 using several pHs as analysis medium, and (**c**) contact angle of a small drop on the hydrogel surface with different pH, together with some representative images.

**Figure 7 polymers-13-04262-f007:**
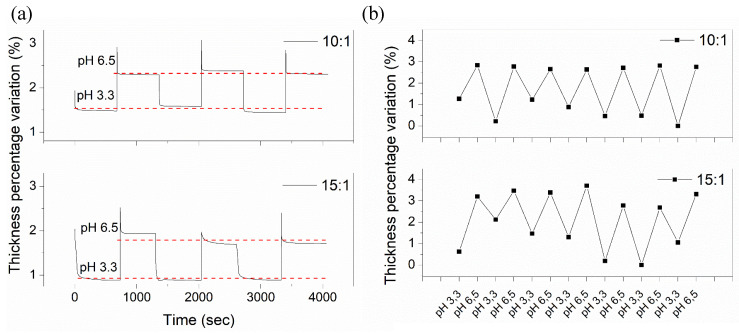
Time dependence of the changes concerning (**a**) QCM thickness and (**b**) ellipsometric measurement upon cycling exposure; Table 3. 3 and 6.5 for two hydrogels (10:1 and 15:1).

**Figure 8 polymers-13-04262-f008:**
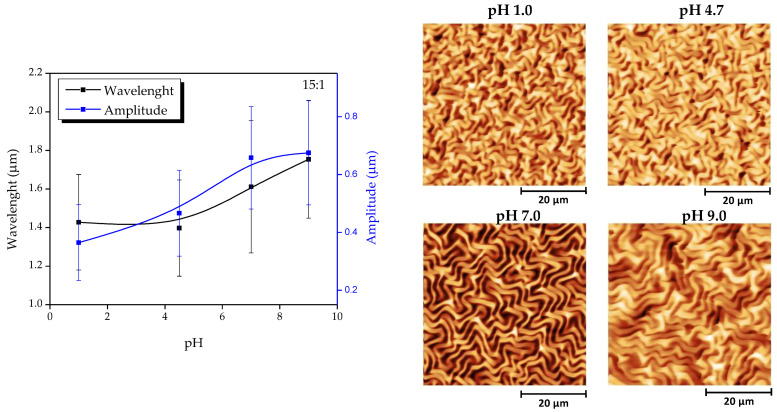
Right: Evolution of the wrinkle characteristics (wavelength and amplitude) for the wrinkled hydrogel 15:1 as a function of Table 1. Obtained at different pH values.

**Figure 9 polymers-13-04262-f009:**
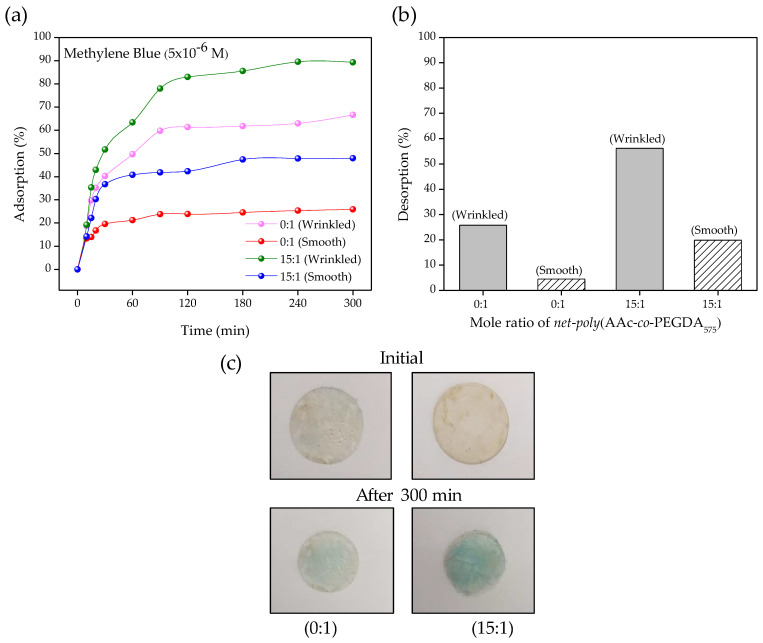
(**a**) MB adsorption percentage, (**b**) MB desorption percentage from smooth and wrinkled hydrogels with a concentration of 0:1 and 15:1, and (**c**) photographs of glasses with different hydrogel concentrations before and after MB adsorption.

**Figure 10 polymers-13-04262-f010:**
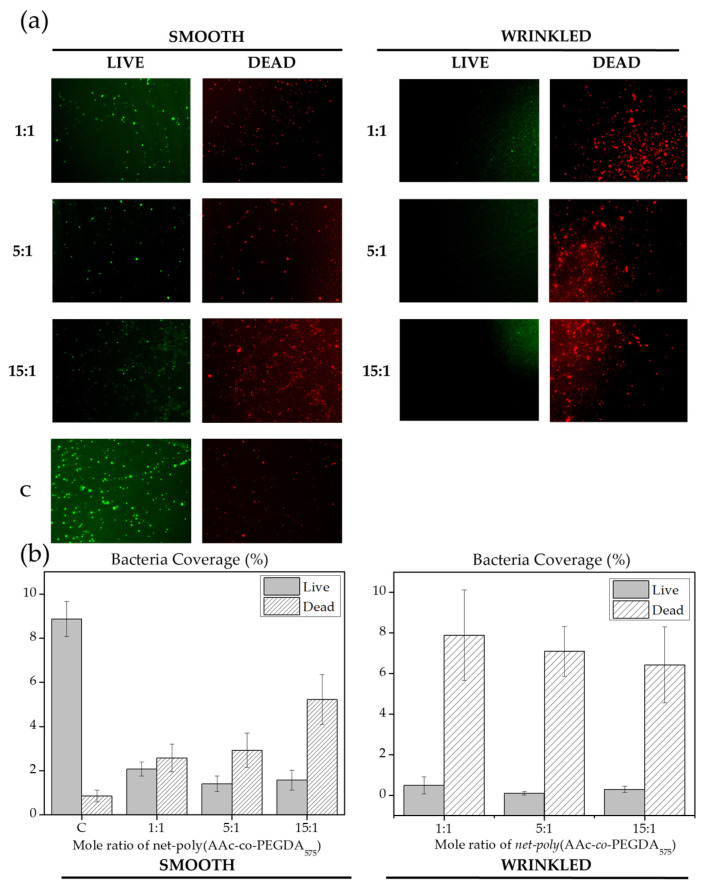
LIVE/DEAD assay for smooth and wrinkled AAc films incubated with *Staphylococcus aureus* bacteria strain: (**a**) live (green) and dead (red) bacteria over wrinkled, smooth, and control (C) films after 48 h of incubation and (**b**) surface area coverage (%) of bacteria culture after ImageJ analysis.

**Figure 11 polymers-13-04262-f011:**
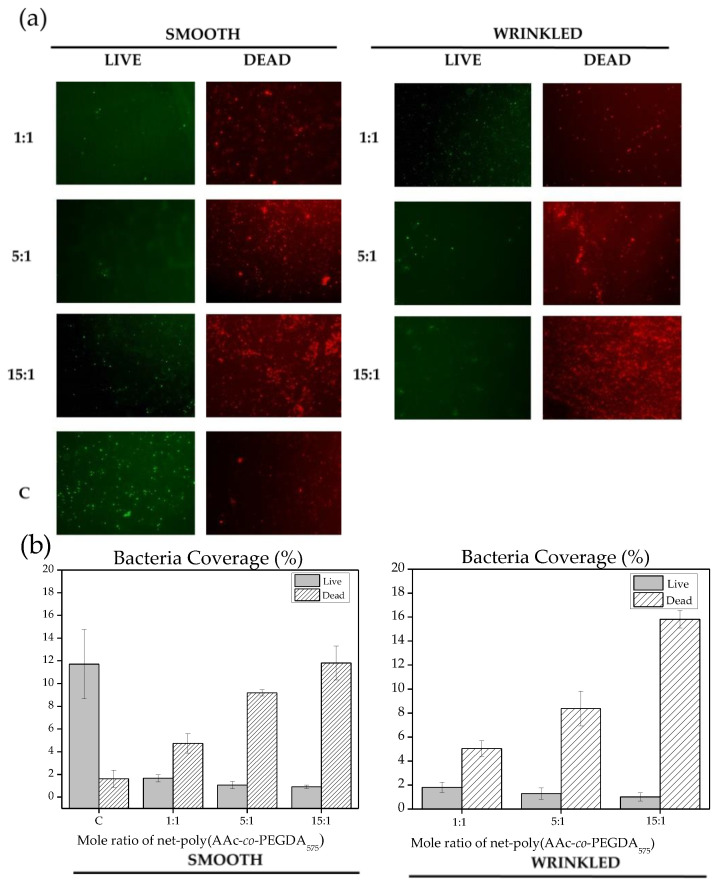
LIVE/DEAD assay for smooth and wrinkled AAc films incubated with *Escherichia coli* bacteria strain: (**a**) live (green) and dead (red) bacteria over wrinkled, smooth, and control (C) films after 48 h of incubation and (**b**) surface area coverage (%) of bacteria culture after ImageJ analysis.

**Table 1 polymers-13-04262-t001:** Reactive amounts used to synthesize the hydrogels *net-poly*(AAc-*co*-PEGDA_575_).

Mole Composition Ratio	AAc (g)	PEGDA_575_ (g)	Irgacure 2959 * (μL)	MilliQ Water (μL)
0:1	0	0.5	6.63	41.7
1:1	0.063		13.3	83.4
5:1	0.313		40.0	250.0
10:1	0.626		72.9	458.3
15:1	0.939		106.0	666.0

* 60 mg of Irgacure 2959 in 300 μL MeOH.

**Table 2 polymers-13-04262-t002:** Real composition of *net-poly*(AAc-*co*-PEGDA_575_).

Mole Ratio (Expected Composition)	Conversion Degree of Each Monomer within the Wrinkled Hydrogel Formed		Real Composition
	AAc	PEGDA_575_	
0:1	0%	81.4%	0:1
1:1	98.7%	94.2%	1.0:0.95
5:1	91.3%	95.0%	4.72:0.95
10:1	99.7%	92.3%	9.93:0.92
15:1	94.3%	89.3%	14.1:0.95

**Table 3 polymers-13-04262-t003:** Thermal studies of the hydrogels *net*-*poly*(AAc-*co*-PEGDA_575_) at different mole ratios for two temperatures (260 °C and 340 °C), with residual mass at 460 °C, and their respective Tg.

Samples	TGA			
	(Weight Loss)			DSC
	260 °C	340 °C	Residual Mass, 460 °C	Tg (°C)
0:1	0.80%	6.00%	3.30%	−29.2
1:1	2.70%	10.00%	4.20%	−15.9
5:1	8.60%	17.50%	6.90%	6.6
10:1	11.50%	23.60%	9.20%	18
15:1	14.00%	27.20%	9.10%	22.7

## Data Availability

The data presented in this study are available on request from the corresponding author.

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
