# Peer review of "Wrinkling on Stimuli-Responsive Functional Polymer Surfaces as a Promising Strategy for the Preparation of Effective Antibacterial/Antibiofouling Surfaces"

_polymers, 2021, doi:10.3390/polym13234262_

Round 1
Reviewer 1 Report
The manuscript reported a kind of smart hydrogel films that could spontaneously form wrinkled surface patterns with pH responsive and antibacterial properties. The hydrogel composites were synthesized by acrylic acid (AAc) as monomer and PEGDA575 as crosslinking agent.
The discussion and analysis about the properties of materials reflect the application value in the field of biomedical and dyes adsorption. The logic in the manuscript is basically clear and the data given in paper can verify the conclusion. This work could be accepted after necessary corrections and the specific suggestion was listed as follows.
- The aims of the work in paper were described in different ways, such as the description in the page 7, line 107-108, and in the page 17, line 598-601, so a clear summary should be given to show the focus and highlights of the study.
- For the first step of the section 2.3, there was a figure describing the effect of the immersion time on the polydopamine coating thickness in air in the Ref. 21. So, there should be the explanation about why the mean measured thickness was inconsistent with reference (10 and 15-20 nm, respectively).
- The manuscript should provide the full name when an abbreviation first appeared, such as PBS (phosphate buffered saline) in the page 6, line 239.
- In Table 2, it would be better if there existed the supporting data for total conversion degree.
- According to the discussion in the manuscript, the generation of free radicals on the surface via frontal vitrification probably could oxidize and polymerize the top layer of the film in the page 5, line 217-219. So, there should also provide the information of smooth hydrogels' chemical composition via 1H-NMR and Raman confocal spectroscopy.
- In the Ref. 15, the research group reported the fabrication of wrinkled hydrogel films with tunable pH sensitivity and used DEAEMA as pH-responsive polymer. So, did the authors compare these hydrogels, such as the coating thicknesses determined by measuring the border's depth profile?
- In somewhere, additional data are needed to prove conclusions. The manuscript provided many data with pH 3.3 and 6.5, and it would be more convincing if there are one or two more sets with other pH, such as in Figure 6 (a). Similarly, in the page 11, line 402-404, it was not persuasive enough to draw the conclusion just through the phenomena at concentrations of 10:1 and 15:1.
- The manuscript did not explain why the dead bacteria's percentage of occupied area would be lower when the acrylic acid content increased on wrinkled surfaces in Figure 9(b).
- The description of Figure 5 (a) was not accurate. Need to carefully revise it.
- The authors could add the following recent review papers into reference which would again increase the interest to smart antibacterial surface readers: Chemical Society Reviews‚ 2021, 50, 8319-8343; Materials Horizons‚ 2021‚8, 1618-1633; J. Mater. Chem. B, 2018,6, 4274-4292.
Author Response
Reviewer #1: The manuscript reported a kind of smart hydrogel films that could spontaneously form wrinkled surface patterns with pH responsive and antibacterial properties. The hydrogel composites were synthesized by acrylic acid (AAc) as monomer and PEGDA575 as crosslinking agent.
The discussion and analysis about the properties of materials reflect the application value in the field of biomedical and dyes adsorption. The logic in the manuscript is basically clear and the data given in paper can verify the conclusion. This work could be accepted after necessary corrections and the specific suggestion was listed as follows.
- The aims of the work in paper were described in different ways, such as the description in the page 7, line 107-108, and in the page 17, line 598-601, so a clear summary should be given to show the focus and highlights of the study.
Answer: The mentioned phrases were modified to express the aim of the work in similar ways; we also added a summary of the work aim in the abstract. We additionally modified the final part of the introduction to highlight the novelty and the features of the materials described. Thanks for your commentary.
- For the first step of the section 2.3, there was a figure describing the effect of the immersion time on the polydopamine coating thickness in air in the Ref. 21. So, there should be the explanation about why the mean measured thickness was inconsistent with reference (10 and 15-20 nm, respectively).
Answer: As you correctly state, in reference [24], Nirasay et al. reported a protocol for functionalizing surfaces with polydopamine SAMs, and, indeed, there is a difference between the reported thicknesses, 10 nm in our case and 15-20 nm in the case of Nirasay et al. We can associate that this difference is mainly related to two factors. First, the substrate that Nirasay et al. used in their study was muscovite mica instead of glass slides with some clear differences in terms of chemical composition. Secondly, in the Nirasay procedure, the authors don't shake the polydopamine solution during deposition time (3 hrs). To corroborate this assumption, we reproduce the protocol reported by Nirasay et al. (using muscovite mica and silicon wafers), and we obtain thicknesses in the same ranges as the reported (between 12.9 nm and 18.6 nm, with an average of 15.7 ± 2.3 nm). We add some of this information in the manuscript in section 2.3.
- The manuscript should provide the full name when an abbreviation first appeared, such as PBS (phosphate buffered saline) in the page 6, line 239.
Answer: Thanks for the commentary; this and other similar inconsistencies about abbreviations were corrected.
- In Table 2, it would be better if there existed the supporting data for total conversion degree.
Answer: The data was already included in the supplementary materials section, and with the finality to clarify the calculations, an explanation was added before Table 2, in section 3.2.
- According to the discussion in the manuscript, the generation of free radicals on the surface via frontal vitrification probably could oxidize and polymerize the top layer of the film in the page 5, line 217-219. So, there should also provide the information of smooth hydrogels' chemical composition via 1H-NMR and Raman confocal spectroscopy.
Answer: Indeed, as you mentioned, frontal vitrification leads to oxidation and polymerization of the top layer; we already report similar systems that present this behavior (DOI: 10.3390/molecules24040751, DOI: 10.1016/j.msec.2018.12.061). To demonstrate it, new analyses on the top polymerization and oxidation were performed and added to the manuscript (section 3.1 and supplementary materials S2-a). Two different samples were analyzed via ATR-FTIR before and after the plasma treatment. Previous to wrinkled pattern formation (before argon plasma irradiation), the samples have fewer carbonyl groups and a lower polymerization degree. After this treatment, an increase in carbonyl vibration and the antisymmetric/symmetric vibration of –CH3 and –CH2– groups were observable, indicating surface oxidation and polymerization of the film top layer.
- In the Ref. 15, the research group reported the fabrication of wrinkled hydrogel films with tunable pH sensitivity and used DEAEMA as pH-responsive polymer. So, did the authors compare these hydrogels, such as the coating thicknesses determined by measuring the border's depth profile?
Answer: In a previous article DOI: 10.1016/j.apsusc.2018.07.022 (mentioned by the referee), our research group explored the pH-sensitiveness of wrinkled hydrogel films based on DEAEMA, HEMA, and PEGDA. In that report, we measure wrinkle width (wavelength) and wrinkle height (amplitude) in samples prepared with different pH (pH 5.4, 7.4, and 8.3) using AFM. These analyses focused on the topography since this will be critical for the final surface properties.
In the present study, we characterized the surface of wrinkled hydrogels films composed by AAc and PEGDA575 with a composition of 15:1 with different pHs (1, 4.5, 7, and 9). The figure below shows the results obtained from the AFM analysis. In the figure at the right is possible to observe some representative images of the samples at variable pHs, and at the left is possible to observe the data obtained from the analysis of the AFM. The results demonstrate that using different pH produces a gradual variation of the wrinkle physical characteristics (wavelength and amplitude). Both parameters depict a slight increase with pHs values above pKa (4.5). This effect can be explained via the deprotonation of the hydroxyl groups of the AAc at alkaline pHs (DOI: 10.1039/C5SM03134F). The environmental pH produces an electrostatic repulsion between the charged groups, leading to the main hydrogel chain stretching, diminishing wrinkle dimensions. When the environmental pH decrease (below pKa), the hydroxyl groups are completely protonated, and the repulsion between the AAc segments has been diminished, thus generating wrinkled patterns with lower amplitude (wrinkle height) and wavelengths (wrinkle width).
We add this information and a new Figure (Figure 8) where we additionally establish a comparison between the two systems in terms of pH response and the size of the wrinkles formed.
- In somewhere, additional data are needed to prove conclusions. The manuscript provided many data with pH 3.3 and 6.5, and it would be more convincing if there are one or two more sets with other pH, such as in Figure 6 (a). Similarly, in the page 11, line 402-404, it was not persuasive enough to draw the conclusion just through the phenomena at concentrations of 10:1 and 15:1.
Answer: With the finality to obtain more data to prove our conclusions, DLS analysis was performed for the different hydrogel compositions (0:1, 1:1, 5:1, 10:1, and 15:1) at two additional pHs (pH 1 and 9). This data was added to Figure 6a and compared with the already obtained data for the hydrogels. Some conclusions about the hydrogel micelle dimensions were reported in section 3.4.
- The manuscript did not explain why the dead bacteria's percentage of occupied area would be lower when the acrylic acid content increased on wrinkled surfaces in Figure 9(b).
Answer: The paragraph covering this topic at the end of the Results section has been rewritten to include the reviewer’s observation. We hypothesize that an additional antifouling or antiadhesive effect for this Gram-positive model could play a role in this scenario, diminishing total bacteria coverage. Although this tendency could be appreciated, no significant differences were detected between 1:1, 5:1, and 15:1 dead S. aureus coverage, showing high efficacy with all samples. Even more, new results obtained from the Gram-negative E. coli model did not reflect this trend. These differences between biological models will be further explored in future experiments, probably including other Gram-positive and negative bacteria strains such as Listeria or Pseudomonas, respectively.
- The description of Figure 5 (a) was not accurate. Need to carefully revise it.
Answer: The description of Figure 5 was modified to clarify the data exposed. Also, section 3.3 was modified to explain carefully the data obtained. Thanks for your commentary.
- The authors could add the following recent review papers into reference which would again increase the interest to smart antibacterial surface readers: Chemical Society Reviews‚ 2021, 50, 8319-8343; Materials Horizons‚ 2021‚8, 1618-1633; J. Mater. Chem. B, 2018,6, 4274-4292.
Answer: Thanks for your commentary; these references were added in the introduction section.
Reviewer 2 Report
The manuscript presents an interesting approach to hydrogel-based biomaterials with structured surface topography and pH responsive behaviour. It is made by copolymerization of a PEG diacrylate and acrylic acid with a special procedure for creating wrinkled surface structures, involving vacuum and Ar plasma exposure.
The materials were thoroughly characterized by a variety of techniques and antibacterial properties were evaluated by a Staph. aureus proliferation experiment.
The experimental work has mostly been carried out well and the results are interesting and relevant for potential biomedical applications. The work fits well to the scope of “polymers” and is in general written well. However, there are a few discrepancies and ambiguities in the description of the experimental procedures and discussion of the results that need to be clarified before publication.
The specific points are listed below:
Chapter 2.3 Methods: The procedure of wrinkled film deposition has been described, but it’s not clear how the smooth films used for comparison were prepared. Have they been directly exposed to UV for 35 min after spin coating, just omitting the vacuum and plasma steps?
Furthermore, its not clear how the wrinkled films for investigations such as NMR, DLS, etc. have been prepared and redissolved for analysis?
Results:
lines 290 and 295: significant bands corresponding to C=C double bonds at 1571 cm-1, and 1460-1480 cm-1 were observed in the Raman spectra. Presence of these bands is in marked contrast to the finding of acrylate conversion rates close to or above 90% as determined by NMR. This discrepancy needs to be discussed in more detail!
lines 387 ff.: DLS investigation: The reported hydrodynamic diameters of several hundred nm indicate that the polymers might be not completely molecularly dissolved and contain larger colloidal particles. Additional information on the polydispersity of the dissolved copolymers would be interesting. Given the high variability of diameters (large error bars in Fig. 6a), the pH induced swelling could ev. be better investigated by repeated DLS measurement of a certain copolymer solution after stepwise change of pH by titration with acid/base.
line 419: The finding that for PAA containing polymers the contact angles are higher (more hydrophobic) at pH values above the pKa is surprising. As also stated in line 419, one would assume more hydrophilic surfaces at higher pH due to the increasing degree of ionization of carboxylic groups. The observed behaviour merits some more detailed discussion and explanation.
lines 441 ff. and 462 ff: QCM and Ellipsometry: It is not clear why different procedures have been applied to deposit the hydrogel films for QCM and Ellipsometry measurements? It was described that already copolymerized hydrogels dissolved in MeOH were coated onto QCM crystals while for Ellipsometry the monomer mixture was spin coated and then UV polymerized on the substrate. These different procedures don’t make sense to me! Actually, the same process described in chapter 2 should be followed for both types of substrates.
Also the results on thickness variation presented in Fig. 7 are not clear. It would be easier to interpret if absolute thickness values in nm are plotted. If plotted as “thickness variation percentage”, why they don´t start with a “thickness variation” of zero? Does the reported thickness variation around 2% mean that for a total thickness around 1500 nm, the absolute measured pH induced thickness changes are in the range of just 30 nm? Given the high film roughness values around 300 nm, its interesting that this can be measured reproducibly and should be discussed in more detail.
line 480: The reported values of 1220 g/mol and 1510 g/mol are probably not the molecular weights of the hydrogel polymers, but the nominal molecular formula weights of the “repeat units”. Polymer MWs will be much higher and would need to be determined by e.g. SEC, static light scattering, or ev. NMR end group analysis.
General: Regarding the pH response of the wrinkled hydrogels, it would be interesting to see AFM images on whether and how the wrinkle patterns are affected by pH changes. Furthermore, a direct comparison of the swelling and pH dependent behaviour of smooth and wrinkled films would be interesting.
Author Response
Reviewer #2: The manuscript presents an interesting approach to hydrogel-based biomaterials with structured surface topography and pH responsive behaviour. It is made by copolymerization of a PEG diacrylate and acrylic acid with a special procedure for creating wrinkled surface structures, involving vacuum and Ar plasma exposure.
The materials were thoroughly characterized by a variety of techniques and antibacterial properties were evaluated by a Staph. aureus proliferation experiment.
The experimental work has mostly been carried out well and the results are interesting and relevant for potential biomedical applications. The work fits well to the scope of “polymers” and is in general written well. However, there are a few discrepancies and ambiguities in the description of the experimental procedures and discussion of the results that need to be clarified before publication.
The specific points are listed below:
- Chapter 2.3 Methods: The procedure of wrinkled film deposition has been described, but it’s not clear how the smooth films used for comparison were prepared. Have they been directly exposed to U.V. for 35 min after spin coating, just omitting the vacuum and plasma steps?.
Answer: Thanks for your commentary. Indeed, we do not include the experimental procedure used to fabricate flat films in the first version of the manuscript, and we apologize for this mistake. We now added this information in section 2.3.4. The flat films were obtained by exposing the samples to 35 min of U.V. light, then exposed to vacuum for two days to extract all of the remnant water of the system and finally irradiated with argon plasma. By generating a stiff layer (dried hydrogel) before argon plasma exposure, we ensure that a wrinkled pattern could not be formed on the top of the material because there is no strain mismatch between the layers inside the film.
- Furthermore, its not clear how the wrinkled films for investigations such as NMR, DLS, etc. have been prepared and redissolved for analysis?.
Answer: Thanks again for your commentary. The experimental procedure used to perform these analyses was added in section 3.2. In a few words, we detached the samples from un-functionalized substrates by immersing them in Mili Q water; then, the sample was dried under vacuum at 60°C and grounded. The obtained powder was used to perform the analysis mentioned before.
Results:
3. lines 290 and 295: significant bands corresponding to C=C double bonds at 1571 cm-1, and 1460-1480 cm-1 were observed in the Raman spectra. Presence of these bands is in marked contrast to the finding of acrylate conversion rates close to or above 90% as determined by NMR. This discrepancy needs to be discussed in more detail!.
Answer: The 1H-NMR data were studied again, and the percentage of PEGDA575 was re-evaluated. We noticed that the sample 5:1 and 10:1 mole ratios showed a higher amount of PEGDA575 that did not react (monomers). These results were written in section 3.2.
- lines 387 ff.: DLS investigation: The reported hydrodynamic diameters of several hundred nm indicate that the polymers might be not completely molecularly dissolved and contain larger colloidal particles. Additional information on the polydispersity of the dissolved copolymers would be interesting. Given the high variability of diameters (large error bars in Fig. 6a), the pH induced swelling could ev. be better investigated by repeated DLS measurement of a certain copolymer solution after stepwise change of pH by titration with acid/base.
Answer: We repeat several of the DLS measurements to determine the reason for the high polydispersity in hydrogel micelle diameter; indeed, we perform new analysis using two different pH (pH 1.0 and 9.0, Figure 6a), and we perform a pH titration for several samples to corroborate our results. As representative data, we depict the titration curve for the sample 15:1 in Figure 6b. The new results were also discussed in section 3.4. We hope these results can contribute to confirming the initial hypothesis.
- line 419: The finding that for PAA containing polymers the contact angles are higher (more hydrophobic) at pH values above the pKa is surprising. As also stated in line 419, one would assume more hydrophilic surfaces at higher pH due to the increasing degree of ionization of carboxylic groups. The observed behaviour merits some more detailed discussion and explanation.
Answer: Thanks for your commentary; indeed, the results have some discrepancies compared to other similar reports (DOI:10.1039/C3PY01339A). We repeated all the data for the analyzed samples to corroborate our results, and the results were included in Figure 6c. Is it possible to observe, those slight contact angle increases were corrected. Thanks for pointing out this mistake.
- lines 441 ff. and 462 ff: QCM and Ellipsometry: It is not clear why different procedures have been applied to deposit the hydrogel films for QCM and Ellipsometry measurements? It was described that already copolymerized hydrogels dissolved in MeOH were coated onto QCM crystals while for Ellipsometry the monomer mixture was spin coated and then U.V. polymerized on the substrate. These different procedures don’t make sense to me! Actually, the same process described in chapter 2 should be followed for both types of substrates.
Answer: We use different substrates for both procedures, but both samples were polymerized after the deposition; we forgot to include this information in the text; thanks for your commentary. Regarding the substrate, we use quartz crystals to measure the changes of the thin films when exposed to pH changes, but it is complicated to obtain the films using spin coating because the quartz crystal is too fragile and tends to break during spin coating (just a microfracture in the crystal could preclude the measuring). On the other hand, silicon wafers are robust enough to perform spin coating and present a reflectancy of more than 99% in the visible range. Part of this information was added to the manuscript. We hope this answer can solve all of your doubts.
- Also the results on thickness variation presented in Fig. 7 are not clear. It would be easier to interpret if absolute thickness values in nm are plotted. If plotted as “thickness variation percentage”, why they don´t start with a “thickness variation” of zero? Does the reported thickness variation around 2% mean that for a total thickness around 1500 nm, the absolute measured pH induced thickness changes are in the range of just 30 nm? Given the high film roughness values around 300 nm, its interesting that this can be measured reproducibly and should be discussed in more detail.
Answer: We choose not to show the results in absolute thickness due to two main reasons. First, and as you correctly stated in the previous question, the deposition methods in both cases differed, resulting in distinct hydrogel film thicknesses. Secondly, in the ellipsometry data, we cannot be entirely sure about the absolute thickness of the sample in this range of thickness. Ellipsometry is a technique that measures slight changes in the refractive index or thickness of thin films, so measuring thicknesses of thick samples could present some discrepancies, but the changes in the thicknesses measured in-situ should not present inconsistencies.
Regarding the low percentages of thickness variation measured, we could justify those slight changes due to the swelled state of the polymeric films. Small conformational changes are easily detectable when the compound is in solution due to the molecules' freedom degrees. In a swelled state, the molecule is restricted; thus, the conformational changes produced due to pH variations are difficult to detect. Part of this information was already added in the manuscript, but new information was included in section 3.4. Thanks for your commentary; it helps us improve the quality of the discussion and analysis of our data
- line 480: The reported values of 1220 g/mol and 1510 g/mol are probably not the molecular weights of the hydrogel polymers, but the nominal molecular formula weights of the “repeat units”. Polymer M.W.s will be much higher and would need to be determined by e.g. SEC, static light scattering, or ev. NMR end group analysis.
Answer: Indeed, you are entirely correct; the values reported were the nominal molecular formula weights, as you stated. To obtain the real polymer molecular weight, we should use those kinds of characterization techniques. The main issue is that, here in Chile, we do not have easy access to that kind of chemical analysis, and due to pandemic restrictions, it becomes much more difficult to perform it. We try to perform a GPC analysis, but we do not have the standards to analyze it.
- General: Regarding the pH response of the wrinkled hydrogels, it would be interesting to see AFM images on whether and how the wrinkle patterns are affected by pH changes. Furthermore, a direct comparison of the swelling and pH dependent behaviour of smooth and wrinkled films would be interesting.
Answer: According to the referee's suggestion, this new version includes a new figure, Figure 8, in which by using AFM, we were able to follow the changes in the wrinkled characteristics as a function of the pH.
Reviewer 3 Report
On request of Polymers, I have revised the manuscript titled “Wrinkling on stimuli-responsive functional polymer surfaces as a promising strategy for the preparation of effective antibacterial/antibiofouling surfaces”, by Carmen M. González-Henríquez and colleges.
The main scope of the present study was to develop effective antibacterial/antibiofouling surfaces. To this end, different pH-responsive hydrogels, based on acrylic acid (AAc) were synthesized, deposited via spin-coating over functionalized glasses, and wrinkles were spontaneously formed by various treatments, to increase cell adherence. Among other characteristics, the pH responsiveness of the materials, the presence of the carboxylic acid functional groups at the interfaces, the adsorption/desorption capacity, and the antibacterial properties were investigated. According to the results, the wrinkled polymer surfaces proved to be capable to establish ionic interactions and to exert excellent antibacterial properties depending on the micro-wrinkling degree and the surface topography (smooth or wrinkled), thus establishing a remarkable improvement of the results obtained for planar (smooth) hydrogel.
General Comments
Studies concerning the development of novel stimuli-responsive bioactive surfaces, as the present one, are welcome since such surfaces currently play a crucial role both in biomedical and tissue engineering applications. The ability of such surfaces to actively change their conformation or physicochemical properties, thus allowing to finely tune their surface attributes, represent a very appealing feature, which can be exploited to produce materials with several different attributes. For this reasons, stimuli-responsive functional polymer surfaces increasingly attract the attention of researchers of several sectors.
Anyway, the strategy herein reported to create wrinkles on functional polymer surfaces is not original, but it is a well-established approach to produce surfaces with several properties and bioactivities, including the antibacterial/antibiofouling ones. In this regard, just Carmen M. González-Henríquez and Juan Rodríguez-Hernández, authors of this study in which “Wrinkling on stimuli-responsive functional polymer surfaces” is presented “as a promising strategy for the preparation of effective antibacterial/antibiofouling surfaces” have already served as Editors and Authors of a book on this topic, also cited in the present work (Book Title: Wrinkled Polymer Surfaces. Book Subtitle: Strategies, Methods and Applications. Editors: C. M. González-Henríquez, Juan Rodríguez-Hernández. DOI, https://doi.org/10.1007/978-3-030-05123-5. Springer Nature Switzerland AG, 2019, Springer, Cham).
So, my major concern, regarding this work is the lack of novelty and originality of the reported strategy, which remarkably lower its scientific relevance. However, the work is well-written, the spontaneous capability of the developed materials to form wrinkles, the performed experiments and the obtained results are interesting. In this regard, I would suggest authors to evidence in the title that in this work stable hydrogel films that could spontaneously form wrinkled surface patterns were fabricated.
Moreover, some other issues (reported below) hamper (for the moment) the publication of this manuscript, and must be addressed.
- In Section 2, subsection titles should be written without indentations. Please, correct the manuscript format accordingly. Moreover, in Section 2.3, please, provide the Steps with a numbering. For example: 2.3.1. Step 1. Substrate functionalization. 2.3.2. Step 2. Preparation of the photosensitive reaction mixture. Etc. Please, write them without indentation.
- According to other articles already published on Polymers, the measure units should be reported as g/L, or mg/mL and not as g L-1 and mg mL-1. In addition, also µl (Table 1) should be changed in µL, as in the footnote of the same Table.
- Please, include the paragraphs “Dye adsorption/desorption tests” and “Antibacterial evaluation” in a new Section, 4. Evaluation tests, as 2.4.1 and 2.4.2 subsections.
- Antibacterial evaluation. The author should provide more details concerning the performed experiments. What was the amount of the inoculum? 0 x 105 CFU/mL or 1.0 x 106 CFU/mL? Please, provide this detail. In addition, the author should perform additional experiments also on a representative strain of the Gram-negative species such as E. coli.
- The authors should perform 1H NMR and Raman analysis also of the two isolate ingredients (AAc and PEGDA575). The obtained spectra should be provided, compared with spectra of the AAc-co-PEGDA575 prepared, and commented. It could be nice if authors, to obtain more useful information by Raman spectral data, process chemometrically the matrix consisting of the spectral data of all the samples. The principal components analysis (PCA) is a useful tool widely applied to interpret the IR, NIR and Raman results.
- Caption of Figure 9 without indentation, please.
- Please, correct the format of the caption of figure S2 in SM.
I suggest Polymers to publish this manuscript only when the abovementioned issues will be addressed. In my opinion, to perform the additional experiments which I required, the authors will need enough time. Therefore, I decided to ask for major revisions.
Author Response
Reviewer #3: On request of Polymers, I have revised the manuscript titled "Wrinkling on stimuli-responsive functional polymer surfaces as a promising strategy for the preparation of effective antibacterial/antibiofouling surfaces", by Carmen M. González-Henríquez and colleges.
The main scope of the present study was to develop effective antibacterial/antibiofouling surfaces. To this end, different pH-responsive hydrogels, based on acrylic acid (AAc) were synthesized, deposited via spin-coating over functionalized glasses, and wrinkles were spontaneously formed by various treatments, to increase cell adherence. Among other characteristics, the pH responsiveness of the materials, the presence of the carboxylic acid functional groups at the interfaces, the adsorption/desorption capacity, and the antibacterial properties were investigated. According to the results, the wrinkled polymer surfaces proved to be capable to establish ionic interactions and to exert excellent antibacterial properties depending on the micro-wrinkling degree and the surface topography (smooth or wrinkled), thus establishing a remarkable improvement of the results obtained for planar (smooth) hydrogel.
General Comments
Studies concerning the development of novel stimuli-responsive bioactive surfaces, as the present one, are welcome since such surfaces currently play a crucial role both in biomedical and tissue engineering applications. The ability of such surfaces to actively change their conformation or physicochemical properties, thus allowing to finely tune their surface attributes, represent a very appealing feature, which can be exploited to produce materials with several different attributes. For this reasons, stimuli-responsive functional polymer surfaces increasingly attract the attention of researchers of several sectors.
Anyway, the strategy herein reported to create wrinkles on functional polymer surfaces is not original, but it is a well-established approach to produce surfaces with several properties and bioactivities, including the antibacterial/antibiofouling ones. In this regard, just Carmen M. González-Henríquez and Juan Rodríguez-Hernández, authors of this study in which "Wrinkling on stimuli-responsive functional polymer surfaces" is presented "as a promising strategy for the preparation of effective antibacterial/antibiofouling surfaces" have already served as Editors and Authors of a book on this topic, also cited in the present work (Book Title: Wrinkled Polymer Surfaces. Book Subtitle: Strategies, Methods and Applications. Editors: C. M. González-Henríquez, Juan Rodríguez-Hernández. DOI, https://doi.org/10.1007/978-3-030-05123-5. Springer Nature Switzerland AG, 2019, Springer, Cham).
So, my major concern, regarding this work is the lack of novelty and originality of the reported strategy, which remarkably lower its scientific relevance. However, the work is well-written, the spontaneous capability of the developed materials to form wrinkles, the performed experiments and the obtained results are interesting. In this regard, I would suggest authors to evidence in the title that in this work stable hydrogel films that could spontaneously form wrinkled surface patterns were fabricated.
Answer: We would like to thank the referee for this appreciation. We want to clarify the context of this work.
We presented in the context of the book as a promising strategy, but most of our previous works were focused on the control of the wrinkling process, analyzing the experimental conditions required for their formation and the characteristics of the wrinkles formed. For instance, we prepared wrinkles with different chemical compositions and even succeeded in preparing hierarchical wrinkled patterns. Also, recently, based on the use of wrinkled hydrogels (rarely explored in the formation of wrinkled patterns), we attempted to employ these patterned surfaces for biorelated applications. More precisely, we explored the cell adhesion and proliferation processes on wrinkled surfaces (for instance, Materials Science and Engineering C, 2019, 103, 109872(1-15)), and started to analyze the potential of these surfaces, when appropriately functionalized, as antimicrobial interfaces. Up to date, we employed fluorinated and, therefore, highly hydrophobic monomers (Materials Science and Engineering C, 2019, 97, 803-812. And Materials Science and Engineering C, 2020, 114, 111031.) for this purpose. Herein, we go one step further and incorporate carboxylic acid functional groups with several essential advantages depicted in the manuscript. First of all, carboxylic acid groups are prone to be functionalized and modified with many other compounds by reaction, such as amines. We evidenced that these groups are available for a chemical reaction (Figure 9). These types of hydrogels can also encapsulate and release positively charged molecules as a function of the environmental pH, and, more importantly, the wrinkled structure significantly improves this capacity. Finally, we evidence the antibacterial activity for two different gram-positive and gram-negative bacteria (Figure 10), where the wrinkled pattern demonstrated to be more effective than the planar counterpart surfaces.
For all these reasons, we believe that this work has the novelty required to be published in a high-quality journal like Polymers. We have, in this version, modified the introduction to highlight these novel aspects attempting to make clear the novelty of the work.
Moreover, some other issues (reported below) hamper (for the moment) the publication of this manuscript, and must be addressed.
- In Section 2, subsection titles should be written without indentations. Please, correct the manuscript format accordingly. Moreover, in Section 2.3, please, provide the Steps with a numbering. For example: 2.3.1. Step 1. Substrate functionalization. 2.3.2. Step 2. Preparation of the photosensitive reaction mixture. Etc. Please, write them without indentation.
Answer: Thanks for the commentary; the changes in the title and subtitle indentation were solved.
- According to other articles already published on Polymers, the measure units should be reported as g/L, or mg/mL and not as g L-1 and mg mL-1. In addition, also µl (Table 1) should be changed in µL, as in the footnote of the same Table.
Answer: Thanks for your commentary, all of the units were changed according to the format requirements.
- Please, include the paragraphs "Dye adsorption/desorption tests" and "Antibacterial evaluation" in a new Section, 4. Evaluation tests, as 2.4.1 and 2.4.2 subsections.
Answer: The proposed changes were made; thanks for your commentary.
- Antibacterial evaluation. The author should provide more details concerning the performed experiments. What was the amount of the inoculum? 1.0 x 105 CFU/mL or 1.0 x 106 CFU/mL? Please, provide this detail. In addition, the author should perform additional experiments also on a representative strain of the Gram-negative species such as E. coli.
Answer: More details about the antibacterial evaluation protocol have been added in the experimental section of the manuscript. In this direct experiment over films surfaces, O.D. readings at 600 nm of bacteria inoculum have been used as a measuring tool to estimate bacteria number previous to seeding over materials. An O.D. of 0.8 was selected for this study, which corresponds to a total number of 6.4 x 108 bacteria/ml, according to Agilent biocalculator for the E. coli model. This bacteria concentration allowed a proper comparison between activated films and control surfaces regarding contact killing ability at 48 hours.
Following the reviewer’s recommendation, a complete evaluation of E. coli behavior has been conducted, obtaining a remarkable antimicrobial activity for this Gram-negative model. A new figure and Results section has been included in the manuscript.
- The authors should perform 1H NMR and Raman analysis also of the two isolate ingredients (AAc and PEGDA575). The obtained spectra should be provided, compared with spectra of the AAc-co-PEGDA575 prepared, and commented. It could be nice if authors, to obtain more useful information by Raman spectral data, process chemometrically the matrix consisting of the spectral data of all the samples. The principal components analysis (PCA) is a useful tool widely applied to interpret the I.R., NIR and Raman results.
Answer: Thank you very much for your comment. The 1H-NMR and Raman results for the PEGDA575 polymer are included in the manuscript, named a 0:1 sample. However, as you mention, the results for the acrylic acid polymer should be considered as a way of comparing all the signals. Accordingly, it was incorporated into the 1H-NMR analysis of the PAAc polymer in Figure S3 (Supplementary materials). Along with this, an FT-IR spectrum of the PAAc polymer was incorporated into the supplemental material (Figure S2-b). In the case of the Raman required, Figure 2 already presented the PEGDA575, the sample, 0:1, that followed the variations due to the incorporation of the acrylic acid. For clarity purposes, a description of the samples measured is included in the caption of Figure 2. Regarded to PCA analysis of Raman, we do not think it is necessary to perform because we can identify all of the signals present in the spectra and relate each one of them with a particular vibration of the molecules analyzed. We do not discard the possibility of performing this type of analysis in the future, but to perform this kind of analysis, sophisticated and expensive software is needed, and we do not have access at this moment. Thanks for your commentary.
- Caption of Figure 9 without indentation, please.
Answer: Thanks for the commentary; figure captions were corrected according to the journal formatting.
- Please, correct the format of the caption of figure S2 in S.M.
Answer: Thanks for the commentary; the caption was corrected.
I suggest Polymers to publish this manuscript only when the abovementioned issues will be addressed. In my opinion, to perform the additional experiments which I required, the authors will need enough time. Therefore, I decided to ask for major revisions.
Round 2
Reviewer 3 Report
Dear Authors,
I have reconsidered the present manuscript which I found significantly improved. By inserting more details or giving more explanations, you have highlighted the originality of the study. You have performed the required additional experiments. Overall the work can now be published on Polymers.